# The Alpha Subunit of Mitochondrial Processing Peptidase Participated in Fertility Restoration in Honglian-CMS Rice

**DOI:** 10.3390/ijms24065442

**Published:** 2023-03-13

**Authors:** Weibo Zhao, Han Geng, Zhiwu Dan, Yafei Zeng, Mingyue Wang, Wuwu Xu, Zhongli Hu, Wenchao Huang

**Affiliations:** 1State Key Laboratory of Hybrid Rice, Wuhan University, Wuhan 430072, China; weibozhao@whu.edu.cn (W.Z.);; 2College of Life Sciences, Wuhan University, Wuhan 430072, China

**Keywords:** cytoplasmic male sterility, fertility restoration, Honglian-CMS rice, mitochondrial processing peptidase, subunit

## Abstract

The cytoplasmic male sterility (CMS) and nuclear-controlled fertility restoration system is a favorable tool for the utilization of heterosis in plant hybrid breeding. Many *restorer-of-fertility* (*Rf*) genes have been characterized in various species over the decades, but more detailed work is needed to investigate the fertility restoration mechanism. Here, we identified an alpha subunit of mitochondrial processing peptidase (MPPA) that is involved in the fertility restoration process in Honglian-CMS rice. MPPA is a mitochondrial localized protein and interacted with the RF6 protein encoded by the *Rf6*. MPPA indirectly interacted with hexokinase 6, namely another partner of RF6, to form a protein complex with the same molecular weight as the mitochondrial F_1_F_0_-ATP synthase in processing the CMS transcript. Loss-of-function of *MPPA* resulted in a defect in pollen fertility, the *mppa*^+/−^ heterozygotes showed semi-sterility phenotype and the accumulation of CMS-associated protein ORFH79, showing restrained processing of the CMS-associated *atp6-OrfH79* in the mutant plant. Taken together, these results threw new light on the process of fertility restoration by investigating the RF6 fertility restoration complex. They also reveal the connections between signal peptide cleavage and the fertility restoration process in Honglian-CMS rice.

## 1. Introduction

CMS is a widespread phenomenon in higher plants characterized by a maternally inherited defect in producing viable pollen grains due to an aberrant chimeric open reading frame (ORF) in the mitochondrial genome [1]. This characteristic of CMS meets the requirement of blocking self-pollination to produce hybrid seeds in many crops, including rice, wheat, and maize [2,3,4]. CMS has been extensively applied in hybrid breeding in crops and the utilization of heterosis, as well as providing an ideal system for learning nucleo-cytoplasmic communications and the coevolution of the nuclear and mitochondrial genome [5,6,7]. In rice, three major types of CMS have been widely acknowledged: the WA (wild abortive)-CMS is sporophytic CMS caused by *WA352*, and the *Rf3* and *Rf4* are the two *Rf* proteins encoding genes responsible for eliminating the influence of WA352 on fertility [8,9,10]; the HL (Honglian)-CMS is gametophytic CMS caused by *atp6-orfH79*, of which sterility could be rescued by *Rf5* and *Rf6* [11,12,13]; the BT (ChinsurahBoro II)-CMS is gametophytic CMS caused by *atp6-orf79*, and the *Rf1a*, *Rf1b*, and *Rf6* proteins contributed to rescuing BT-CMS [14,15,16]. Intriguingly, the *Rf* proteins alone are not sufficient for rescuing the sterile phenotype in many cases, they work with other proteins for precise cleavage of the CMS-associated transcripts in mitochondria to rescue sterility, suggesting the complexity of the mechanisms in fertility restoration [12,13,17,18].

The *Rf* proteins are encoded by the nuclear genome, most of which are members of the pentatricopeptide repeat (PPR) protein family with mitochondrial signal peptide in the *N*-terminus [19,20]. Upon translocation, the *N*-terminal signal peptides in the precursors of the *Rf* proteins are removed by mitochondrial processing peptidase (MPP) and then targeted to the corresponding sub-compartments within mitochondria [21]. The MPP is a heterodimeric metalloendopeptidase that is essential for the maturation of proteins imported to mitochondria [21]. In plants, the MPP is usually integrated into and substitutes the core I and core II subunits of the mitochondrial inner membrane-anchored cytochrome *bc*_1_ complex (complex III), without forfeiting the cleavage or electron transfer activity [22,23]. Thus, the dual roles of the MPP give the first example of a connection between electron transfer and the mitochondrial protein import events in plants [21,24,25]. The α- and β-subunits of the MPP are highly conserved with similar architecture in plants and animals [25]. The glycine-rich and negatively charged carboxyl terminus of the α-subunit is involved in the substrate recognition and binding [26,27,28], while the β-subunit holds catalytic activity that is responsible for peptide bond cleavage [29]. The proteins imported into mitochondria are processed from pre-mature to mature states in one or multiple steps and MPP functions in cleavage events as the first step [30]. However, little is known about the posttranslational processing of the *Rf* proteins, which is a necessary step for their maturation before participating in fertility restoration.

The aberrant chimeric mRNA *atp6-orfH79* encoded a toxic protein, ORFH79, that causes pollen sterility in Honglian-CMS rice [31]. We have previously found that the P-type PPR protein, RF6, worked together with hexokinase 6 (OsHXK6) and other uncharacterized factors to rescue the sterile phenotype of Honglian-CMS rice via forming a macromolecular protein complex in mitochondria to cleave the aberrant chimeric transcript [13]. To obtain a better understanding of the fertility restoration mechanism via RF6, we managed to screen an uncharacterized protein that interacts with RF6 specifically in mitochondria. Further study indicated that the uncharacterized protein was predicted to be the alpha subunit of mitochondrial processing peptidase, fundamentally influential for the correct assembly of the fertility restoration complex. In summary, these results demonstrated that the MPPA is an indispensable part of the RF6 fertility restoration complex, implying the link between the signal peptide cleavage apparatus and fertility restoration events.

## 2. Results

### 2.1. MPPA Interacts with RF6 In Vivo and In Vitro

To investigate the fertility restoration process of Honglian-CMS rice, we performed a screening assay against the yeast two-hybrid library constructed previously [13], using the RF6 as bait. A predicted alpha subunit of mitochondrial processing peptidase was identified to potentially interact with RF6 from screening (Figure 1A). The MPPA was classified as a subunit of Zn^2+^-dependent peptidase (Figure 1B), and qRT-PCR data indicated that the MPPA expressed in various rice tissues was similar to that of RF6, implying the physical interaction potential between these two proteins in rice (Figure 1C).

To validate the interaction between MPPA and RF6, we expressed and purified the recombined Histidine (His) tagged MPPA and glutathione S-transferase (GST)-RF6 fusion proteins from *E. coli* and performed an in vitro pull-down assay (Appendix A). We found that His-MPPA directly interacted with GST-RF6, as detected by immunoblotting (Figure 1D). Furthermore, we performed an in vivo immunoprecipitation assay using the anti-RF6 antibody. We noticed that the MPPA protein was detected with the anti-MPPA antibody in the precipitation, but not in the IgG control, confirming the interaction of RF6 with MPPA in vivo (Figure 1E). Since the RF6 fertility restoration complex was recruited to cleave the CMS-associated mRNA, the presence of RNA might mediate the interaction between MPPA and RF6 in vivo. However, RNase A treatment did not disperse the interaction between RF6 and MPPA in a parallel experiment (Figure 1E).

Sequence analysis indicated that the MPPA consists of two domains, M16 and M16C. To determine which domain of MPPA mediated the interaction with RF6, we generated a series of truncated versions of MPPA (Figure 1B, Appendix A), and test the interactions with RF6 via the yeast two-hybrid system. The M16 and M16C domains of the MPPA interact with RF6, as well as the truncated proteins with these two domains (Appendix A). Together, these results indicated that MPPA interacted with RF6 directly in an RNA-independent manner.

### 2.2. MPPA Is Localized to Mitochondria

The subunits of mitochondrial processing peptidase function mainly in the protein complex of the inner mitochondrial membrane, participating in electron transfer and signal peptide removal processes [32]. Previously, we found that RF6 was targeted to mitochondria to rescue Honglian-CMS rice [13]. Therefore, MPPA is expected to be a mitochondrial protein since it cooperated with RF6 directly. To verify the subcellular localization of MPPA, we transiently expressed the full-length MPPA coding sequence fused with green fluorescence protein (GFP) driven by *CaMV 35S* promoter in rice protoplast (Appendix A). GFP signals of MPPA-GFP were observed and well merged with mitochondria, and no GFP signal was detected in other cellular compartments (Figure 2B, Appendix A), indicating that MPPA was specifically targeted to mitochondria.

### 2.3. Dysfunction of MPPA Leads to Defect in Pollen Viability

To test whether MPPA is required for the fertility restoration process involved in *Rf6*, we generated knockout lines of *MPPA* using the CRISPR/Cas9 method in the *Rf6* near-isogenic line (*Rf6*-NIL) background. The *Rf6*-NIL represented an ideal material for investigating the mechanism in CMS/*Rf* system. The heterozygous knockout lines of *mppa*^+/−^ showed a semi-sterile phenotype compared to the wild-type (WT) plant, as indicated by pollen staining with 1% I_2_-KI (Figure 3A–C, Appendix A). Furthermore, we found that the germination rate of the *mppa*^+/−^ was lower than that of the WT plants in the pollen germination assay in vitro (Figure 3D–G). Nevertheless, the pollen tube elongated normally in the stigma of both transgenic and WT plants, suggesting that pollens containing *mppa* were inviable, while the others with the MPPA were sufficient for fertilization in *mppa*^+/−^ heterozygous plants (Figure 3I–L). This inference was in concurrence with the observation that no homozygote was obtained from the offspring of *mppa*^+/−^ heterozygotes (Table 1). Instead, the progenies of heterozygotes exhibited a 1:1 segregation ratio of WT to heterozygous plants (Table 1). Moreover, the protein level of MPPA was reduced in the pollens of transgenic plants, leading to the accumulation of ORFH79 in pollens of *mppa*^+/−^ (Figure 4, Appendix A).

Next, we conducted the reciprocal crosses between the wild-type rice and the *mppa*^+/−^ heterozygotes to confirm that the mutation in *MPPA* led to sterile phenotype and pollen inviability. As expected, when the different *mppa*^+/−^ heterozygotes served as male parents and the wild-type plant as the female parents, the progenies showed the same phenotype as the wild-type plant with 100% fertility without separation of character. While using the wild-type plant as male parents and the three lines of *mppa*^+/−^ heterozygotes as female parents, the progenies exhibited a 1:1 segregation ratio of WT to heterozygous plants (Table 2). Collectively, these results indicated that *MPPA* was indispensable for fertility restoration in CMS rice, and dysfunction of *MPPA* disrupted the RF6 fertility restoration complex without impact on female gamete development.

### 2.4. MPPA Interacts with OsHXK6 Indirectly

To further explore the role of MPPA in the fertility restoration process, we tested the association of MPPA with another component of the RF6 fertility restoration complex, OsHXK6. Thus, we expressed and purified the recombined maltose-binding protein (MBP) tagged MPPA protein and His-OsHXK6 from the prokaryotic expression system. However, the recombined MBP-MPPA did not pull down with His-OsHXK6, suggesting that MPPA might not interact with OsHXK6 directly (Appendix A). We speculated that an intermediate component might be responsible for the association between these two proteins. Thus, a co-immunoprecipitation assay was applied to test the association of MPPA and OsHXK6 in vivo. We found that OsHXK6 was precipitated with MPPA, as detected by the anti-OsHXK6 antibody, and the association between them was RNA-independent (Figure 1C). Together, these results indicated that MPPA was indirectly associated with OsHXK6 in an RNA-independent pattern.

### 2.5. MPPA Is Conserved in Eukaryote

Sequence analysis indicated that MPPA harbored two domains, the peptidase M16 in the amino terminus and M16C in the carboxyl terminus. Then we performed an alignment analysis using the full-length MPPA as a query against the National Center for Biotechnology Information database. Homologs of MPPA are distributed in various eukaryotes, including animals and plants. Phylogenetic analysis indicates that MPPA and its homologs were divided into two clusters from the 18 typical species. The homologs of MPPA in animals predominantly encoded the mitochondrial cytochrome bc1 complex subunit 2, while plant MPPA belongs to the peptidase family M16 (Figure 5). Interestingly, subunits of MPP from plants integrated into the protein complex of the respiratory chain substituted the core components of the cytochrome bc1 complex [21]. Together, these data indicated that MPPA was highly conserved in plants and animals, suggesting that the core components of the cytochrome *bc*_1_ complex might be the relics of the subunits of MPP.

### 2.6. MPPA Is Indispensable for the RF6 Fertility Restoration Complex

To verify the biological function of MPPA in the fertility restoration complex, we performed a blue-native PAGE analysis with mitochondria lysates from *Rf6*-NIL and YTA. Immunoblotting indicated that both MPPA and RF6 were presented in the protein complex of the same molecular weight in the *Rf6*-NIL, rather than in the sterile rice (Figure 6A). Moreover, combined with the immunoblotting evidence and the Coomassie Brilliant Blue staining result, we found that the protein complex exhibited the same molecular weight as the mitochondrial F_1_F_0_-ATP synthase (Complex V). This observation also demonstrated that MPPA localized to mitochondria, as shown in Figure 2B. These results indicated that MPPA was an indispensable member of the RF6 fertility restoration complex, and RF6 was required in recruiting the protein complex.

## 3. Discussion

The CMS and fertility restoration system contributed to the application of heterosis in hybrid breeding, thus dramatically increasing crop production to ensure food security [1,33]. Many *Rf* genes have been characterized and applied in hybrid breeding, but there is still some scope for explaining the underlying mechanism of fertility restoration [34]. Previously, we cloned the restorer of fertility gene *Rf6* and another component of the fertility restoration complex, OsHXK6 [13]. *Rf6* encoded a pentatricopeptide repeat protein RF6, together with OsHXK6, forming a protein complex to cleave the CMS-causing transcript *atp6-orfH79*. However, neither RF6 nor OsHXK6 bind to *atp6-orfH79*, suggesting that other proteins should be involved in the RNA cleavage in the fertility restoration machinery [13].

In the present work, we identified a new component of the RF6 fertility restoration complex, MPPA. MPPA is an alpha subunit of mitochondrial processing peptidase (MPP), indicating the association between protein complex recruitment and signal peptide removal events. In vivo and in vitro data demonstrated that MPPA interacts with RF6 directly in an RNA-independent manner. Meanwhile, we detected the association of MPPA with OsHXK6. The indirect interaction between MPPA and OsHXK6 suggested the distance in spatial position within the macromolecular protein complex for these two members. Hence, we proposed an updated version of the working model of the RF6 fertility restoration complex, where MPPA was involved in the signal peptide removal process after capturing the pre-mature RF6 protein. After signal peptide cleavage, the RF6-interacted MPPA helped RF6 recruit other components to assemble the protein complex. The MPPA was the core component of the inner membrane-anchored cytochrome *bc*_1_ complex, as a consequence, assisted the RF6 fertility restoration complex in anchoring into the mitochondrial inner membrane. Then the CMS-associated mRNA was cleaved, following the rescuing of the sterile phenotype (Figure 6B). However, the proteins involved in RNA binding and cleavage in the RF6 fertility restoration complex remain vague. More work needs to be done to identify the RNA binding proteins and RNase, and this will help in understanding the fertility restoration process in HL-CMS rice, as well as for BT- and WA-CMS, since the RNA cleavage factors are also unknown for *atp6-orf79* and *WA352* [10,35,36].

Transgenic experiments showed that the heterozygous mutant lines of *mppa*^+/−^ in the *Rf6*-NIL background exhibited a semi-sterile phenotype. Furthermore, we obtained no homozygous mutants of *mppa^−/−^* from the selfed heterozygotes plants. Combined with the reciprocal cross results, we concluded that pollens containing mutated *MPPA* were inviable. This observation was in accordance with the characteristics of gametophytic sterility that the pollen was sterile when the fertility restoration machinery was destructed. These results further confirmed that loss-of-function of *MPPA* leads to the dysfunction of the RF6 fertility restoration complex in Honglian-CMS rice. Moreover, we detected that MPPA and RF6 were presented in a protein complex of the same molecular weight by blue native-PAGE, and the two-dimensional electrophoresis and immunoblotting experiments would provide strong evidence for this conclusion, but we failed to obtain these data due to technological reasons. These data indicated that MPPA participated in the fertility restoration process via interacting with RF6 protein.

The removal of signal peptides within premature proteins transported into mitochondria via mitochondrial processing peptidase strictly depends on the intact structure of both subunits of MPP [22,23]. Plant MPPs are usually integrated into the respiration complex III to assist electron transportation beyond peptidase activity [37]. As a core component of the MPP, MPPA consequently probably helps RF6 anchor into the inner membrane of mitochondria. Additionally, the higher expression level of *MPPA* compared with *Rf6* suggested the multi-functions of *MPPA* beyond involvement in the fertility process. Thus, MPPA might mediate crosstalk among the cytochrome *bc*_1_ complex, mitochondrial processing peptidase, and the RF6 fertility restoration complex. Moreover, we found that MPPA clustered into different branches in *japonica* and *indica* via phylogenetic analysis using the 3K Rice Genomes Project data. *MPPA* is distributed more widely in the *indica* subspecies, suggesting the different evolutionary directions of MPPA in the *japonica* and *indica* subspecies (Appendix A). In this study, we reported that the alpha subunit of the MPP participated in the fertility restoration process, and more work needs to take to explore the precise function of MPPA. Together, we characterized a new member of the RF6 fertility restoration complex and brought out a more detailed work into the CMS and nuclear-controlled fertility restoration system.

## 4. Materials and Methods

### 4.1. Plant Materials and Growth Conditions

*Rf6*-NIL was used as the WT and all mutant background materials. *Rf6*-NIL was derived from repeated backcrossing of 9311 to YTA (Yuetai A, HL-CMS line). The *mppa*^+/−^ mutant plants were generated via the CRISPR/Cas9 system in an approach described previously [38]. The target of Cas9 was within the first exon and intron of *MPPA* (Target sequence CCTCGGGGCCATCAAGGTCAGCC) and the *mppa*^+/−^ indicated the heterozygous mutant while the *mppa^−/−^* indicated the homozygous mutant. The genomic region flanks the CRISPR target site of MPPA were amplified and subjected to Sanger sequencing for mutant certification, and three lines of the mutants were selected for further analysis. All of the plant materials were cultivated at the experimental field of Wuhan University in Wuhan, China.

### 4.2. Plasmid Construction

All the primers used in this study are listed in Appendix A. The *Rf6* constructs for protein expression and yeast two-hybrid assay were generated as described previously [13]. The *MPPA* constructs for the yeast two-hybrid, subcellular localization, and protein expression were generated by One-Step Rapid Cloning Kit (Yeasen Biotechnology). Briefly, full-length MPPA amplified from cDNA was subcloned into the NdeI site of pGBKT7 and pGADT7 for verifying interaction with RF6 or self-activation test with primers MPPA-AD-F and MPPA-AD-R. For subcellular localization analysis, full-length MPPA was subcloned into the BamHI site of the HBT-sGFP (S65T)-NOS vector. Full-length MPPA was subcloned into the BamHI site of pET-32a and pMAL-c2x to generate the fusion protein constructs with Histidine (His) or Maltose binding protein (MBP) tag, respectively.

### 4.3. Bioinformatic Analysis

Protein sequence analysis of MPPA was accomplished with Motif Scan (https://myhits.sib.swiss/cgi-bin/motif_scan, accessed on 1 July 2019) software and Transmembrane analysis was conducted via TMHMM Server v.2.0 (http://www.cbs.dtu.dk/services/TMHMM/, accessed on 23 January 2023) software. The full-length amino acid sequence of MPPA was put into BLASTP for homologous analysis, multiple protein sequence alignments were performed with Clustal X and the phylogenetic tree was built by MEGA7 based on the neighbor-joining method. Furthermore, we analyzed the *MPPA* sequence in the 3K Rice Genomes Project database (https://snp-seek.irri.org/_snp.zul;jsessionid=ABF789EB036AE614AC6DB95D1A080CFF, accessed on 4 September 2020) and generated the phylogenetic tree of MPPA in rice.

### 4.4. Gene Expression Analysis by RT-qPCR

Total RNA was extracted from various rice tissues from *Rf6*-NIL, including leaf (L), 0.5 cm panicle (P0.5), 1.5 cm panicle (P1.5), 3 cm panicle (P3), 5 cm panicle (P5), 8 cm panicle (P8), 12 cm panicle (P12), stem (S), internode (I), and root (R), using TRIzol reagent (ambion). Total RNA was treated with DNase I (Thermo Fisher Scientific, Waltham, MA, USA) before the first-strand cDNA was reverse transcribed. cDNA was synthesized using the M-MLV reverse transcriptase Kit (Thermo Fisher Scientific) with oligo (dT) primer. qPCR was performed with Hieff qPCR SYBR Green Master Mix (No Rox) (Yeasen Biotechnology) on a LightCycler 480 platform (Roche Applied Science) with three biological replications, each with three technical repeats. The relative expression levels were calculated by the 2^−ΔΔCT^ method, based on *actin* (GenBank: KX302608) internal control.

### 4.5. Subcellular Localization of MPPA

The full-length coding sequence of *MPPA* was subcloned into HBT95-sGFP (S65T)-NOS transient expression vector and then transfected into rice protoplast as described, with minor modifications [39]. The empty vector was used as a control. Mitochondria were stained with MitoTracker Red and observed under the confocal microscope (Leica TCS SP8) at an excitation wavelength of 644 nm, and signals of GFP were observed at an excitation wavelength of 488 nm.

### 4.6. Pollen Tube Germination Assay

For pollen tube germination in stigma, spikelet was collected on a time scale from 10 min to 150 min and fixed for 24 h in FAA fixative before dehydration. Then the pollen tubes were observed on an inverted fluorescence microscope (Leica DMi8) after being stained with 0.1% aniline blue dye. In vitro pollen tube germination assay was performed as described with minor modifications [40]. Briefly, the freshly opened flowers were collected and put into the germination medium, and the pollen was collected after vortex for 1 min, then the pollen was germinated at 28 °C for 0.5~1 h. The germinated tubes were observed using the 10× lens on an inverted microscope (Olympus IX53).

### 4.7. Protein Expression and In Vitro Pull-Down Assay

The His-MPPA plasmid was transformed into *E. coli* BL21 (DE3) for recombinant protein expression that was induced by 0.25 mM isopropyl-β-D-thiogalactoside (IPTG) at an OD_600nm_ of 0.6. After growing for 12 h at 16 °C, the cells were harvested and washed twice with 10 mM PBS (10 mM Na_2_HPO_4_, 2 mM KH_2_PO_4_, 137 mM NaCl, 2.7 mM KCl, pH 7.4) solution, then subjected to sonication. Before being applied to the Ni-NTA agarose resin column (Yeasen Biotechnology), the lysate was centrifuged and filtrated with a 0.45 μm polytetrafluoroethylene (PTFE) membrane. The loaded Ni-NTA column was washed with 10 mM Imidazole and then His-MPPA recombinant protein was eluted by buffer containing 250 mM Imidazole, 50 mM Na_2_HPO_4_, 300 mM NaCl, pH 8.0. The MBP-MPPA recombinant protein was purified the same way as His-MPPA with the amylose resin column (NEB), and the loaded column was washed with 0.5 mM maltose, then MBP-MPPA recombinant protein was eluted by buffer containing 10 mM maltose, 20 mM Tris-HCl, 200 mM NaCl, 1 mM EDTA, pH 7.4. The GST-RF6 and His-OsHXK6 were purified as described previously [13,41]. The pull-down assay was carried out according to the protocol [13]. Immunoblotting was conducted with anti-His (1:5000 dilution, MBL), anti-GST (1:5000 dilution, EnoGene, New York, NY, USA), and anti-MBP (1:5000 dilution, MBL).

### 4.8. Coimmunoprecipitation (Co-IP) Assay

The cell lysate from *Rf6*-NIL was incubated with protein A/G magnetic beads (MCE) that coupled to the anti-RF6 antibody at 4 °C for 3 h. The magnetic beads were washed with Co-IP buffer (20 mM Tris-HCl, 150 mM NaCl, 1 mM EDTA·Na_2_, 0.2% *v*/*v* NP-40, 0.1% Cocktail (ThermoFisher Scientific, Waltham, MA, USA), pH 7.4) five times, then the proteins were denatured in SDS-loading buffer at 98 °C for 10 min before separating on 10% SDS-PAGE. Immunoblotting was performed with anti-RF6 (1:1000 dilution), anti-MPPA (1:1000 dilution), and anti-OsHXK6 (1:5000 dilution) antiserum.

### 4.9. Blue-Native PAGE

The BN-PAGE was performed as described previously [42], with moderate modifications. Briefly, mitochondria were extracted from callus cells, then lysed on ice for 0.5 h with buffer containing 5.625 mM 6-Aminohexanoic acid, 37.5 mM Bis-Tis, 0.375 mM EDTA·Na_2_·H_2_O, 10% n-Dodecyl β-D-maltoside and 0.1% protease inhibitor Cock-tail, pH 7.0. The mitochondrial lysate was centrifuged at 20,000× *g* for 20 min to collect the supernatant before separating on the 4~16% continuous gradient gel. The samples run on the gradient gel at 100 V for 4 h and 500 V for 8 h, 4 °C.

## 5. Conclusions

In this study, we identified a novel member of the RF6 fertility restoration complex from yeast two-hybrid screening and confirmed the interaction of MPPA with RF6 in vitro and in vivo. Transgenic experiments indicated that the mutation of *MPPA* resulted in pollen abortion in *Rf6*-NIL, further confirming that *MPPA* is required in the fertility restoration process in Honglian-CMS rice. MPPA is predicted as the alpha subunit of the mitochondrial processing peptidase, consequently probably helping RF6 anchor into the inner membrane of mitochondria.

## Figures and Tables

**Figure 1 ijms-24-05442-f001:**
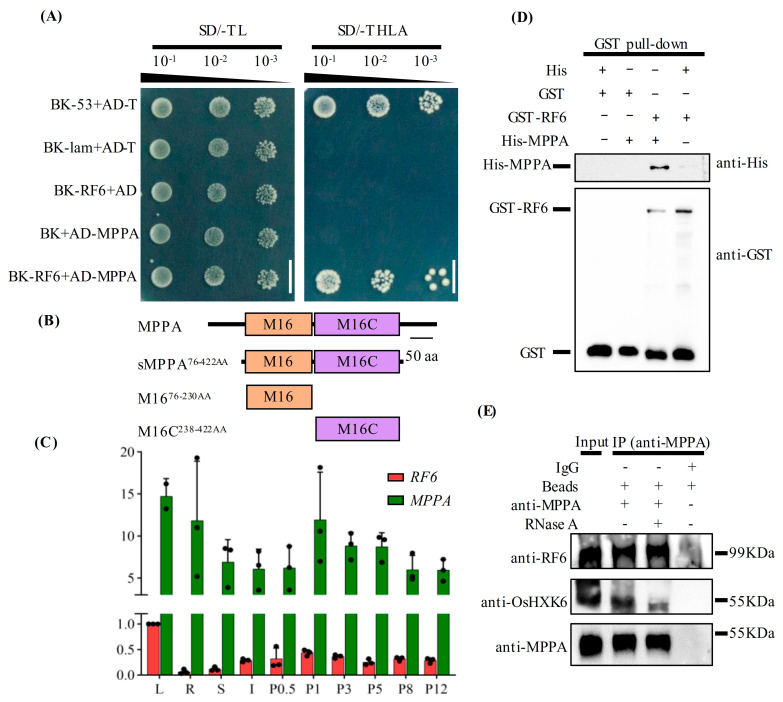
MPPA interacts with RF6. (**A**) For the yeast two-hybrid assay, full-length cDNA of MPPA was cloned into the pGADT7 vector and the RF6 coding sequence was cloned into the pGBKT7 vector, then co-transformed into the yeast strain AH109 to test the interactions. Bar = 10 mm. (**B**) MPPA is a predicted mitochondrial processing peptidase subunit, and the major domains were indicated: sMPPA, two domains with link region (residues 76–422); M16, M16 domain (residues 76–230); M16C, M16C domain (residues 238–422). (**C**) *MPPA* is expressed universally in different rice tissues similar to that of *Rf6*. L, leaf. R, root. S, stem. I, internode. P, panicle. Numbers after the letters indicate different panicle lengths, unit: centimeter (cm). Data are presented as mean ± S.D. from three biological repeats. *Actin* was used as an internal control. (**D**) In vitro GST pull-down using recombinant GST-RF6 and His-MPPA proteins. Glutathione S-transferase (GST) and Histidine (His) were used as control. GST pull-down was performed with glutathione resin and Western blots were carried out with GST and His antibodies at a dilution of 1:5000. (**E**) Co-immunoprecipitation assay indicated that MPPA was associated with RF6 and OsHXK6 in vivo in an RNA-independent manner. IgG was used as negative control and immunoblotting was performed with anti-MPPA (1:1000), anti-OsHXK6 (1:5000), and anti-RF6 (1:1000) antiserum.

**Figure 2 ijms-24-05442-f002:**
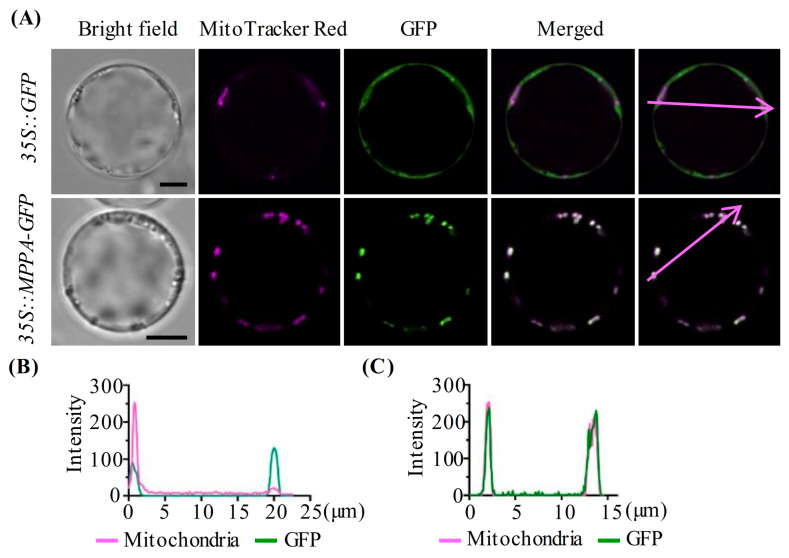
MPPA is localized to mitochondria. (**A**) Subcellular localization analysis of MPPA in rice protoplast. Magenta, MitoTracker Red, living cell fluorescent dye. GFP, green fluorescence protein. Bar = 5 μm. (**B**,**C**) Quantification of co-localization of mitochondria and MPPA (**C**) and the GFP control (**B**).

**Figure 3 ijms-24-05442-f003:**
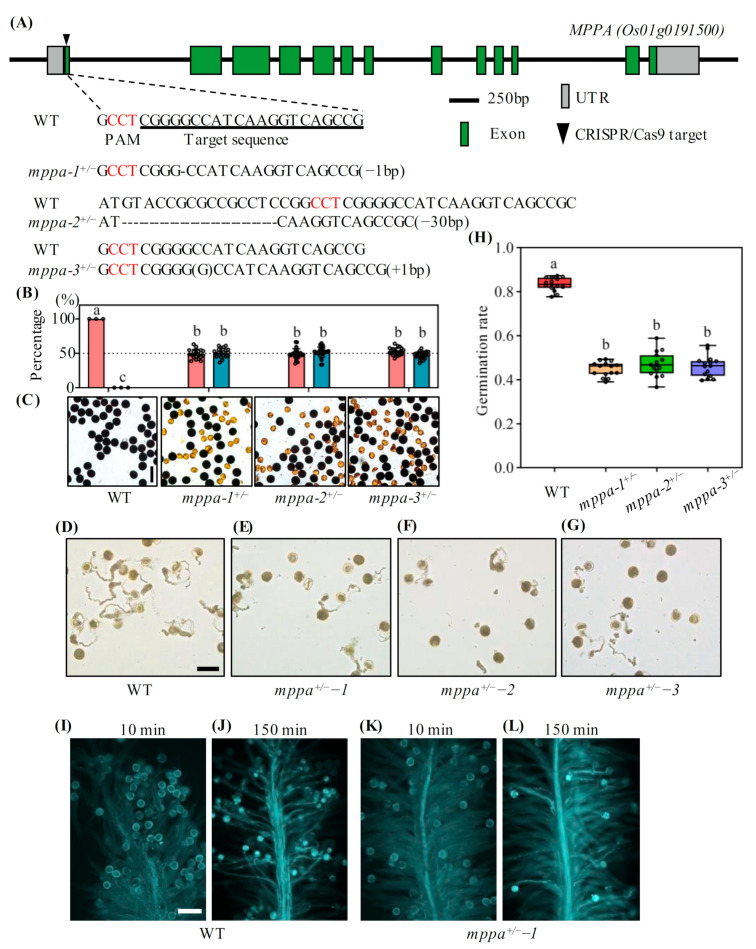
Dysfunction of *MPPA* leads to defects in pollen fertility. (**A**) Schematic diagram of *MPPA* (upper panel) and the genotypes of corresponding CRISPR/Cas9 mutant lines (lower panel) for the following study. The letters in red indicated the PAM site. (**B**) Percentage of fertile and sterile pollens in WT and three *mppa*^+/−^ heterozygotes. The same letter indicates a lack of significant difference using the Tukey–Kramer multiple comparison test with one-way analysis of variance (ANOVA) between groups, *p* < 0.05 was considered significant. (**C**) 1% I_2_-KI staining of pollen grain from WT and three *mppa*^+/−^ heterozygous mutant lines. Bar = 50 μm. (**D**–**G**) In vitro pollen tube germination of wild type rice (**D**) and *mppa*^+/−^ heterozygotes (**E**–**G**). Bar = 50 μm. (**H**) The germination rate of wild-type rice and different *mppa*^+/−^ heterozygotes. Fifteen individuals were counted for each line. Different letters indicate significant differences (*p* < 0.0001) according to Tukey’s test (one-way ANOVA between groups). (**I**–**L**) Pollen eggs implanted in the stigma after selfing. The stigmas were stained with 0.1% aniline blue dye. Bar = 100 μm.

**Figure 4 ijms-24-05442-f004:**
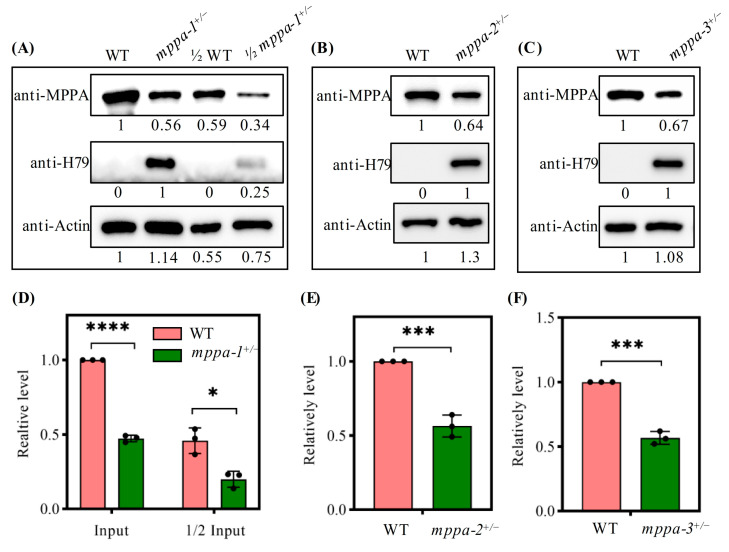
Semi-quantitative analysis of MPPA protein levels in the anther of WT and heterozygotes. (**A**–**C**) Total protein samples were prepared from the anther of the wild-type rice and heterozygous mutants during the heading stage. Loading quantity of lanes 3 and 4 were half of lanes 1 and 2 in (**A**). Three biological repeats were performed for each mutant line. (**D**–**F**) Quantification of the MPPA abundance in pollens of the wild-type rice and heterozygous mutants, data present mean ± SD from three independent biological repeats, Student *t*-test was applied, * *p* <0.05, *** *p* < 0.001, **** *p* < 0.0001. Actin was used as a loading control, H79 indicates the toxic protein, and the numbers calculated by Image J indicated the gray value of immunoblotting. Western blotting was performed with anti-MPPA and anti-H79 at a dilution of 1:1000, and anti-Actin at 1:5000.

**Figure 5 ijms-24-05442-f005:**
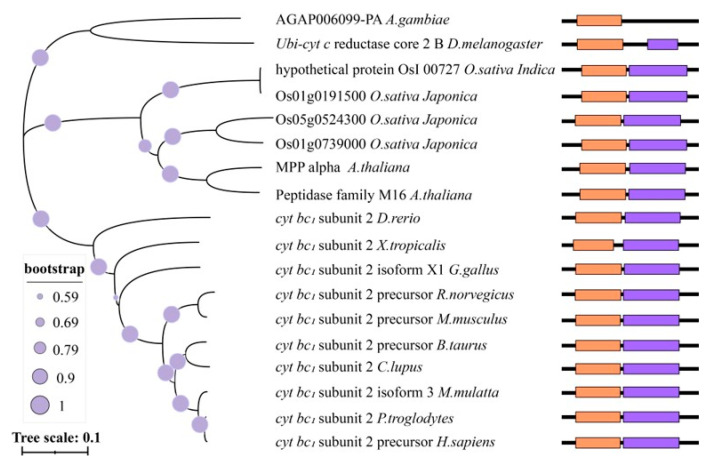
MPPA is conserved in the eukaryote. The full-length amino acid of MPPA and its 17 homologs were aligned by Cluster X, then the phylogenetic tree was constructed based on the neighbor-joining method with MEGA v.7.0. The p-distance model was applied with a bootstrap of 1000 replicates and the corresponding support values for each node was labeled. The right panel showed the conserved domain of MPPA and its homologs, orange rectangles indicate the M16 domain and the purple ones represent the M16C domain.

**Figure 6 ijms-24-05442-f006:**
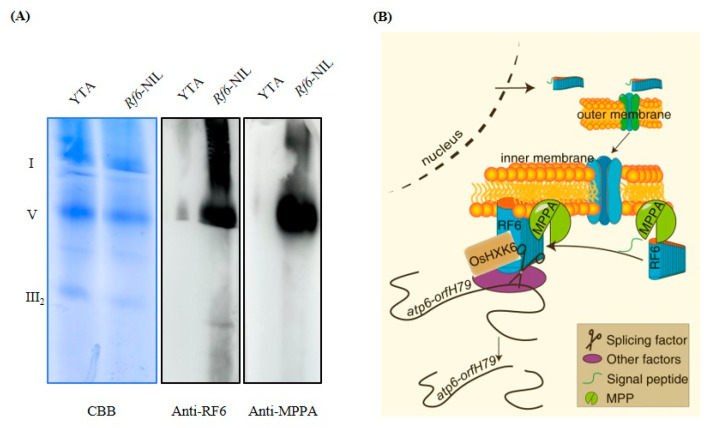
MPPA is indispensable for the RF6 fertility restoration complex. (**A**) The blue-native PAGE (BN-PAGE) analysis of the mitochondrial protein complex. Anti-RF6 and anti-MPPA were used for detecting the RF6 fertility restoration complex in YTA and *Rf6*-NIL, respectively. (**B**) An updated version of the working model of the RF6 fertility restoration complex. Upon translocation, the signal peptide of RF6 protein was removed by the MPP complex, and the alpha subunit (MPPA) of the MPP complex interacted with the RF6 to help it anchor into the inner membrane of mitochondria. The CMS-associated *atp6-orfH79* was cleaved after the assembly of the RF6 fertility restoration complex. Then the fertility was restored in Honglian-CMS rice.

**Table 1 ijms-24-05442-t001:** *mppa*^+/−^ T_3_ Segregation.

Genotype	Num	Ratio	χ^2^
WT: *mppa-1*^+/−^	25:17	1:1	1.524 *
WT: *mppa-2*^+/−^	39:41	1:1	0.05 *
WT: *mppa-3*^+/−^	21:21	1:1	0 *

“*” The chi-square test was adopted. *p* > 0.05, no significant difference.

**Table 2 ijms-24-05442-t002:** Reciprocal crosses between *mppa*^+/−^ and WT.

♀ × ♂	F_1_	χ^2^
*mppa-1*^+/−^ × WT	WT: *mppa-1*^+/−^ = 39:41	0.05 *
WT × *mppa-1*^+/−^	WT: *mppa-1*^+/−^ = 50:0	
*mppa-2*^+/−^ × WT	WT: *mppa-2*^+/−^ = 28:21	1 *
WT × *mppa-2*^+/−^	WT: *mppa-2*^+/−^ = 35:0	
*mppa-3*^+/−^ × WT	WT: *mppa-3*^+/−^ = 33:37	0.23 *
WT × *mppa-3*^+/−^	WT: *mppa-3*^+/−^ = 42:0	

“*” The chi-square test was adopted. *p* > 0.05, no significant difference.

## Data Availability

The sequences of genes and proteins mentioned in our study are available for download from the public database mentioned above.

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
