# Peer review of "The Alpha Subunit of Mitochondrial Processing Peptidase Participated in Fertility Restoration in Honglian-CMS Rice"

_ijms, 2023, doi:10.3390/ijms24065442_

Round 1

Reviewer 1 Report

Overall, it is a well written manuscript with good presentation & discussion of the results. The querry is:

(1) During the discussion (and in Fig. 6B), it is indicated that there could other players or factors involved in the RF6 complex, if so what could be the hypothesis? Does this conclusion mean that the working model proposed is still work in progress, explain.

(2) How does this working model proposed in Honglian CMS compared to others like i.e., Wild abortive or Boro...

Author Response

Point 1: During the discussion (and in Fig. 6B), it is indicated that there could other players or factors involved in the RF6 complex, if so what could be the hypothesis? Does this conclusion mean that the working model proposed is still work in progress, explain.

Response 1: We thank the reviewer’s constructive comments. As indicated in Fig.6A, the RF6 complex share the same molecular weight as the mitochondrial complex V (~580kDa), and in previous work, we found that the RF6 and OsHXK6 protein do not cleave the atp6-orfH79 transcript, as well as the MPPA (no enzyme activity domain was detected), so there must be other proteins participated in RNA binding and cleavage. It is intriguing why such a big protein complex specifically processes a mitochondrial chimeric RNA, so we are now working on identifying the other components of this protein complex and whether this protein complex or some components of it have other functions, as well as how this protein complex is recruited in mitochondria. At present, we have identified several other components, transgenic data has been collected, the molecular biological experiments are undergoing.

Point 2: How does this working model proposed in Honglian CMS compared to others like i.e., Wild abortive or Boro..

Response 2: We thank the reviewer’s constructive comments. In WA-CMS rice, the Rf4 decreased the accumulation of WA352 transcript and the Rf3 probably regulated the expression of WA352 to restore fertility, but whether the Rf4 protein played as a participator or directly in degrading the WA352 remains exclusive. In BT-CMS rice, RF1A participated in the cleavage of the atp6-orf79 transcript and RF1B participated in the degradation of it, but neither of the RF1A and RF1B has RNase activity. So, there should be other proteins that participated in the fertility restoration process in WA- and BT-CMS rice, but whether there are protein complexes that exist is still unknown. Moreover, the Rf6 is functional in restoring BT-CMS, thus, the proposed working model of Honglian-CMS will be of help in understanding how the nuclear-encoded genes regulate mitochondrial gene expression and how the mitochondrial chimeric RNA is being processed.

Reviewer 2 Report

Dr. Huang and colleagues conducted a fascinating study in which they identified an alpha subunit of mitochondrial processing peptidase (MPPA) that is involved in the fertility restoration process in Honglian-CMS rice. The manuscript is well-presented, with an impressive study design and well-documented results. The statistical and bioinformatics analyses are thorough and reliable. This study is highly relevant to the field, and I strongly recommend that it be considered for publication in the journal.

Author Response

Point 1: Dr. Huang and colleagues conducted a fascinating study in which they identified an alpha subunit of mitochondrial processing peptidase (MPPA) that is involved in the fertility restoration process in Honglian-CMS rice. The manuscript is well-presented, with an impressive study design and well-documented results. The statistical and bioinformatics analyses are thorough and reliable. This study is highly relevant to the field, and I strongly recommend that it be considered for publication in the journal.

Response 1: We appreciate the reviewer’s commendation on our work and constructive comments.

Reviewer 3 Report

Authors submitted manuscript entitled “The alpha subunit of mitochondrial processing peptidase participated in fertility restoration in Honglian-CMS rice” to IJMS. In this manuscript,  They identified an alpha subunit of mitochondrial processing peptidase (MPPA), involved in the fertility restoration process in Honglian-CMS rice. They found that MPPA is a mitochondrial localized protein which interacts with RF6 protein encoded by the Rf6. The loss-of-function mutants of MPPA show defect in pollen fertility, the mppa+/- heterozygotes showed semi-sterility phenotype and the accumulation of CMS-associated protein ORFH79, showing restrained processing of the CMS-associated atp6-OrfH79 in the mutant plant. This work is good, and manuscript is written very well. I have few comments and suggestions.

Authors should describe the mutation types for the mentioned mutants in main text (homo or hetero). Why authors used heterozygous mutants in their study?

Authors should provide the information about the selected target sites for Cas9 in methodology.

How about pollen viability of mutants?

L334; provide the locus ID of actin gene used as an internal control.

Discussion lacks comprehensive speculation of results in relevance to previous results and recent findings. Moreover, provide the statements about research gaps and possible future studies.

Author Response

Point 1: Authors should describe the mutation types for the mentioned mutants in main text (homo or hetero). Why authors used heterozygous mutants in their study?

Response 1: We thank the reviewer’s kind suggestion. We used the “mppa+/-” for the heterozygous mutant and “mppa-/-” for the homozygous mutant in this manuscript and added the mutation type for the mentioned mutants in the Method part.

The mutation of MPPA leads to pollen sterility, so, the pollens containing mppa were sterile and the mutated gene can only pass on to the next generation via the female parent. Therefore, no homozygous mutants were obtained.  

Point 2: Authors should provide the information about the selected target sites for Cas9 in methodology.

Response 2: We thank the reviewer’s kind suggestion. We have added the information about the selected target sites for Cas9 in Method part, and the target of Cas9 was within the first exon and intron of MPPA (Target sequence CCTCGGGGCCATCAAGGTCAGCC).

Point 3: How about pollen viability of mutants?

Response 3: 50% pollen of the heterozygous mutants could not be stained by 1% I2-KI, indicating that no starch was accumulated. Moreover, the pollen tube elongation assay indicated that the pollen germination rate of the heterozygous mutants decreased significantly, indicating that the pollen viability was reduced in the mutants.

Point 4: L334; provide the locus ID of actin gene used as an internal control.

Response 4: We thank the reviewer’s kind suggestion, and the GenBank accession number (KX302608) was added in the Method part.

Point 5: Discussion lacks comprehensive speculation of results in relevance to previous results and recent findings. Moreover, provide the statements about research gaps and possible future studies.

Response 5: We appreciate the reviewer’s critical comments, we have revised the Discussion part. In this manuscript, we identified MPPA as a new component of the RF6 fertility restoration complex, indicating the association between protein complex recruitment and signal peptide removal events. However, the proteins involved in RNA binding and cleavage in the RF6 fertility restoration complex remain vague, more work need to take to identify the RNA binding proteins and RNase, and this will help in understanding the fertility restoration process in HL-CMS rice, as well as for BT- and WA-CMS, since the RNA cleavage factors are also unknown for atp6-orf79 and WA352.
